# Design of ultra-swollen lipidic mesophases for the crystallization of membrane proteins with large extracellular domains

Alexandru Zabara[1], Josephine Tse Yin Chong [1], Isabelle Martiel [2], Laura Stark[3], Brett A. Cromer [4,5], Chiara Speziale[1], Calum John Drummond [4] & Raffaele Mezzenga [1]

In meso crystallization of membrane proteins from lipidic mesophases is central to protein structural biology but limited to membrane proteins with small extracellular domains (ECDs), comparable to the water channels (3–5 nm) of the mesophase. Here we present a strategy expanding the scope of in meso crystallization to membrane proteins with very large ECDs. We combine monoacylglycerols and phospholipids to design thermodynamically stable ultra-swollen bicontinuous cubic phases of double-gyroid (Ia3d), double-diamond (Pn3m), and double-primitive (Im3m) space groups, with water channels five times larger than traditional lipidic mesophases, and showing re-entrant behavior upon increasing hydration, of sequences Ia3d→Pn3m→Ia3d and Pn3m→Im3m→Pn3m, unknown in lipid self-assembly. We use these mesophases to crystallize membrane proteins with ECDs inaccessible to conventional in meso crystallization, demonstrating the methodology on the Gloeobacter ligand-gated ion channel (GLIC) protein, and show substantial modulation of packing, molecular contacts and activation state of the ensued proteins crystals, illuminating a general strategy in protein structural biology.

[1] Department of Health Sciences and Technology, ETH Zurich, Schmelzbergstrasse 9 LFO E23, 8092 Zürich, Switzerland. [2] Swiss Light Source, Paul Scherrer Institute, 5232 Villigen, PSI, Switzerland. [3] Faculty of Medicine and Dentistry, University of Western Australia, 35 Stirling Highway Perth, Perth, WA 6009, Australia. [4] School of Science, College of Science Engineering and Health RMIT University 124 La Trobe Street, Melbourne, VIC 3000, Australia. [5] Department of Chemistry and Biotechnology, Swinburne University of Technology, John Street, Hawthorn, VIC 3122, Australia. Alexandru Zabara, Josephine Yin Tse Chong contributed equally to this work. Correspondence and requests for materials should be addressed to R.M. (email: raffaele.mezzenga@hest.ethz.ch)

Membrane proteins play a critical role in mediating cellular processes, as they reside in the lipid bilayer of biological membranes and they are responsible for communication between the intracellular and extracellular environments. Understanding their function is of paramount importance in designing and developing drugs and pharmaceuticals targeting disorders or diseases caused by their malfunction or change in activity.

Crystallizing membrane proteins is a challenging task, particularly for those with large extracellular domains (ECDs), yet, this provides the foundation for membrane protein structural biology. Specifically, crystallization is to date the only viable approach for resolving the complex three-dimensional protein structures and therefore deciphering the underlying mechanisms of their intercellular interactions.

Since its inception in 1996[1], the in meso membrane protein crystallization has become a revolutionary technique leading to the resolution of ~360 membrane protein structures in the Protein Data Bank (PDB). This alternative approach relies on a lipid, often a monoacylglycerol, which once combined with water spontaneously self-assembles into a biomimetic artificial membrane capable of providing a "native-like" mesophase for membrane protein reconstitution and, upon additional precipitants, crystal nucleation and growth[2,3]. Among the greatest recent successes of the in meso method, it is noteworthy the resolution of the structure and therefore the understanding of the mechanism of action of the highly relevant class of G-Protein coupled receptors [4–6].

The tools developed to date for the lipidic cubic phase (LCP) crystallization methods span from automating and miniaturizing processes in sample preparation for high-throughput imaging and screening precipitant conditions[3,7] to the assessment of various host lipids capable of crystallizing membrane proteins under different conditions (e.g., pH, temperatures) [8–14].

In spite of recent progress made toward understanding the mechanistic changes that drive crystal formation within the lipidic bilayer[15,16] as well as enhancing the widespread use of in meso crystallization[7–9], a major limiting factor hindering crystal formation has remained the relatively small size of the lipidic mesophase aqueous domains. Typically LCPs are characterized by two sets of interpenetrating and non-interconnected water channels with a diameter of 3–5 nm, separated by a three-dimensional lipid membrane percolating through space. This geometric constraint prevents the reconstitution of membrane proteins with large extracellular or intracellular domains and restricts the use of the lipidic host matrices to membrane proteins with small hydrophilic domains.

Attempts to overcome this major structural limitation have been done in and beyond the context of membrane proteins crystallization. Cherezov et al[17] incorporated a series of additives, mainly small amphiphiles, leading to an increase of the mesophase lattice parameter up to 40% before a transition to a highly disordered sponge phase was observed and used this system as host for membrane protein crystallization. Other formulations have been proposed with varying degrees of success to swell the host lipidic mesophases while preserving the bicontinuous cubic phase symmetry, including sugar esters[18], surfactants (e.g., octyl glucoside)[19,20], lipids (e.g., diglycerol monooleate (DGMO)[21], cholesterol[22,23]), and phospholipids (e.g., soybean phosphatidylcholine[20], dioleoyl phosphatidylserine (DOPS)[23,24], dioleoyl phosphatidylglycerol (DOPG)[24], distearoyl phosphatidylglycerol (DSPG)[25,26]).

Electrostatic swelling of the mesophase via addition of charged lipids is a promising tool to generate thermodynamically stable swollen cubic phases. Engblom et al.[25] have used an anionic phospholipid, DSPG, to swell the monoolein (MO)/water

mesophase at room temperature up to a maximum lattice parameter of 268 Å and were able to obtain Im3m, Ia3d, and Pn3m cubic phases at different lipid–phospholipid–water ratios. They also observed that the electrostatic swelling effect from the anionic phospholipids, used in the ternary MO–phospholipid–water systems, allows for substantially more water (~70% w/w) to be contained within the swollen cubic phase than the binary MO–water system (~40% w/w)[25,26]. By mixing charged lipids and cholesterol as membrane-stiffening agent, the largest thermodynamically stable and structurally ordered cubic phases were obtained by Barringa et al.[24] and Tyler et al.[23] achieving swollen Im3m cubic phases with a maximum lattice parameter approximately four times larger than the classical (MO)/water sytem[24]. However, the largest swelling was observed at high temperature (i.e., 35–45 °C[23] and 55 °C[24]) and the only symmetry observed was the primitive Im3m cubic phase, both factors being unsuitable for in meso membrane protein crystallization, which typically requires lower temperatures and Pn3m symmetry. By combining the electrostatic swelling effects with epitaxial growth from capillary walls, very recently Kim et al.[27] reported super swollen mesophases of Im3m, Ia3d, and Pn3m symmetries with lattice parameters up to of 68.4 nm, and the unusual coexistence of double-gyroid symmetries with excess water[27]. Nonetheless, these conditions were reported far from equilibrium, and the meta-stability of the system reduces its applicability for membrane protein crystallization, which may require weeks for crystals to nucleate and growth: indeed authors did not consider it for this specific use and to date, in meso crystallization of membrane proteins from thermodynamically stable ultra-swollen LCP is still to be achieved.

Here we present a system based on anionic phospholipid (DSPG) with monoacylglycerol monopalmitolein (MP) leading to thermodynamically stable ultra-swollen cubic phases of Ia3d, Pn3m, and Im3m symmetries. We study the phase diagram of these systems and show a re-entrant behavior of the Ia3d and Pn3m symmetries upon increasing hydration level, which is responsible for a swelling of the lattice parameters and water channels up to five-fold compared to typical lipidic mesophases. We exploit the thermodynamic stable nature of these systems to crystallize a representative membrane protein with large ECDs, the *Gloeobacter violaceus* ligand-gated ion channel (GLIC), otherwise inaccessible to the classical in meso crystallization techniques (Fig. 1), and we show how the crystallization in meso of this protein leads to significantly improved packing of the proteins within the crystals and a differently observed space group compared to all the deposited structures of the same protein obtained by vapor diffusion crystallization, opening a promising strategy for crystallization of challenging membrane proteins.

## Results

**Fine-tuning the design of ultra-swollen LCP systems.** Our starting point in the formulation of the LCP is the selection of a monoacylglycerol, MP, different from the traditional MO for the relatively shorter hydrophobic tail (C16 vs. C18 for GMO), but known for its higher maximum hydration point, as well as the capacity to form cubic phases with larger structural parameters than those found in the MO–water system[28]. To this system, DSPG was used as an electrostatic swelling lipid[25]. Although MP supports the presence of cholesterol in the ensued mesophases, with and without DSPG (Supplementary Figure 4 & 5), cholesterol was not added here, since the membrane proteins targeted in the present work do not require its presence for reconstitution and crystallization.

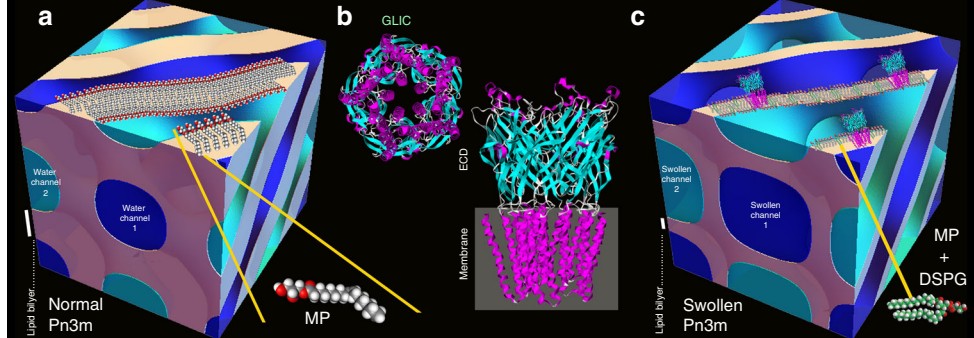

**Fig. 1** Normal vs. swollen mesophases and the GLIC protein structure. Schematic illustrations of **a** normal Pn3m cubic mesophase composed of MP:water, **b** GLIC protein structure, and **c** in meso crystallization of GLIC protein in a highly swollen Pn3m cubic mesophase composed of DSPG:MP:water. Note the difference in size of the lipid bilayer scale bar, which is 30 Å, on **a** and **c** panels

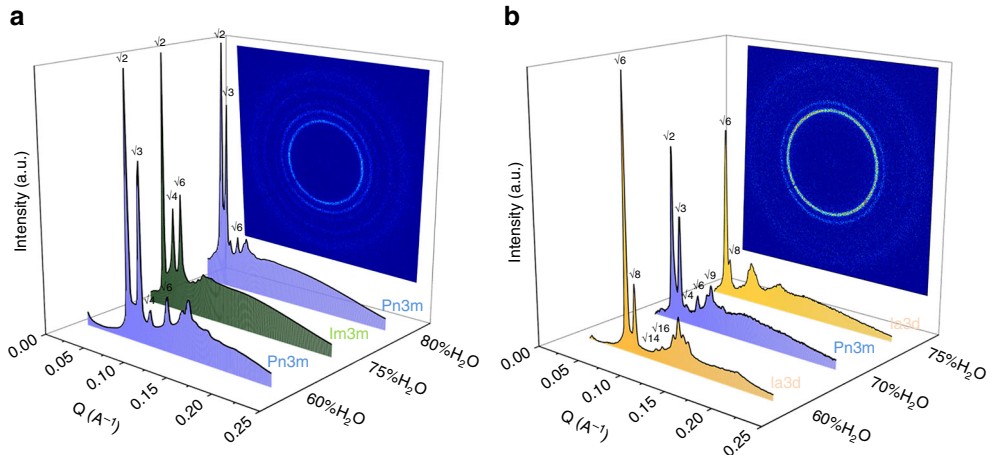

**Fig. 2** SAXS profiles of the DSPG–MP–water systems. One-dimensional SAXS spectra of scattered intensities vs. scattering vector **Q** for **a** 5 wt% DSPG/MP with 60, 75, and 80% water, and **b** 8 wt% DSPG/MP with 60, 70, and 75% water at 20 °C. Re-entrant phase behavior is shown here with the different-colored spectra representing the DSPG/MP/water system with Pn3m (blue), Im3m (green), and Ia3d (orange) cubic phases

Before having an in-depth analysis of the MP–DSPG–water system, we wanted to determine how the addition of DSPG to the MP–water system influences the maximum hydration point of the lipid mixture as well as the structural parameters of the formed mesophases. Synchrotron small-angle X-ray scattering (SAXS) analysis of the resulting mesophases revealed a substantial swelling compared to that of MO–water systems[25], and allowed the assessment that varying amounts of DSPG lead to different bicontinuous cubic symmetries (Fig. 2 and Supplementary Figure 1). More precisely, addition of 3, 5, and 8 wt% of DSPG resulted in the formation of highly swollen-primitive Im3m, double diamond Pn3m, and double gyroid Ia3d cubic phases (Fig. 2) at different levels of hydration. This opens up the possibility of designing suitable LCP symmetries for membrane protein crystallization via minor changes in lipid composition.

Most importantly, SAXS analysis at 80% hydration revealed the room temperature (20 °C) formation of bulk phase, highly swollen bicontinuous cubic phases of both double-gyroid and double-diamond symmetries that far supersede all binary monoacylglycerol–water and most of the previously reported swollen mesophases in terms of structural parameters. Respectively, we observed an Ia3d cubic phase with a lattice parameter of 525 Å and a water channel diameter of 226 Å, and a Pn3m cubic phase with a lattice parameter of 301 Å and a water channel diameter of 204 Å (approximately five times larger than the classical MO–water cubic phase used for membrane protein

crystallization). Their thermodynamically stable nature at 20 °C makes them the ideal hosting matrices for membrane proteins with large ECDs.

Further increasing the amount of DSPG (up to 10 wt%) resulted in a coexistence of phases (Ia3d and $L_\alpha$) at maximum hydration. This could be tentatively explained by localized hydrated domains of self-assembled phospholipids ($L_\alpha$) within the swollen cubic matrix, once the maximum doping capacity of the system is reached (Supplementary Figure 1).

**Phase diagram and re-entrant behavior of MP–DSPG–water.** Analysis of the phase diagram (Fig. 3a) revealed that the DSPG–MP–water system was able to retain 10% more water than the DSPG–MO–water system, as expected a priori from our choice of the specific monoacylglyerol used. The ability of the lipid system to retain larger amounts of water directly modulates the maximum attainable structural parameters and therefore plays a key role in the use of the system for membrane protein crystallization.

Along with the capacity to retain more water, the DSPG–MP–water system revealed another surprising feature in lipid mesophase order-to-order transitions. More specifically, for two distinct lipid compositions containing different amounts of added phospholipid (5 or 8 wt%), a re-entrant behavior of the respective swollen bicontinuous cubic phase at the higher hydration levels was consistently observed. For example, in the

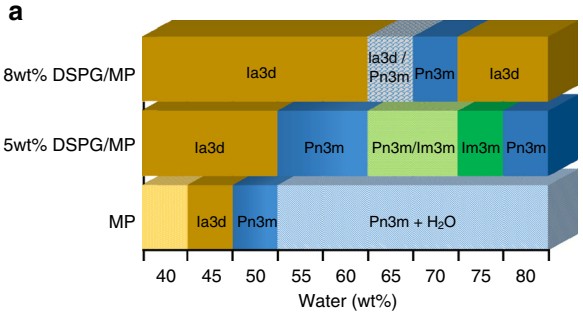

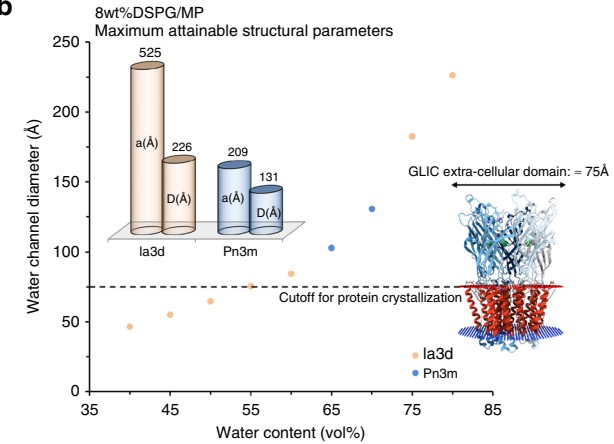

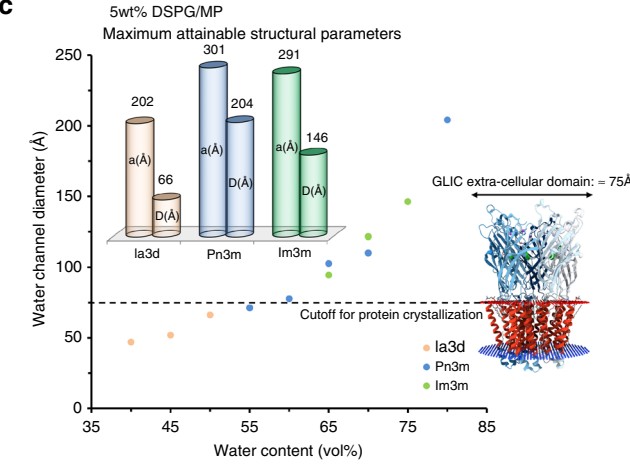

**Fig. 3** Phase diagram and water channel sizes of the DSPG–MP–water system. **a** Phase diagram of swollen systems using 5 and 8 wt% DSPG/MP/water compared with the normal MP/water system, and water channel diameter size as a function of the total amount of water available in the **b** 8 wt% and **c** 5 wt% DSPG/MP/water systems for the different cubic mesophases considered. Maximum attainable water channel structural parameters are shown in the top left section of the plot. The cubic mesophases Pn3m, Im3m, and Ia3d are represented using blue, green, and orange colors, respectively. GLIC protein extracellular domain size is shown on the right side of the plot

case of the 5 wt% DSPG/MP-water system, a double-diamond Pn3m cubic phase is initially observed at 55–60% w/w hydration. Upon increasing the water content in the system to 65–75% w/w $H_2O$, the phase transitions to a primitive Im3m cubic phase. A further increase in water content, which would normally drive the system toward a coexistence with excess water, induces a second order-to-order transition back to the highly swollen double-diamond Pn3m cubic phase at 80% w/w hydration (Fig. 3).

Similarly, the 8 wt% DSPG/MP–water system transitions from an initial double-gyroid Ia3d cubic phase at 60% w/w water content to a Pn3m cubic at 65–70% w/w hydration, before returning to the highly swollen Ia3d symmetry at higher hydration levels (75–80% w/w water). Although re-entrant behavior is typically observed when competing interactions are at play, at the moment, the driving forces behind these unusual transitions remain unclear and mandate deeper future investigations.

Figure 3b, c show the evolution of the water channel size as a function of the total amount of water available in the system, at a fixed temperature of 20 °C for the different mesophases considered in the MP:DSPG:water systems containing 8 wt% and 5 wt% phospholipid, respectively (for the calculation of the channel radii see supporting information and supporting Eq. 1 to 2a–c). The re-entrant phase behavior and the large increase in structural parameters can both be readily observed. More importantly, this allows setting the thresholds for the use of the lipid matrices for the encapsulation and crystallization of membrane proteins with large ECDs (e.g., GLIC). By comparing the size of the LCP water channel with the diameter of the extracellular domain of the model protein GLIC (~75 Å), we can determine the minimum level of hydration needed in the MP–DSPG system in order to obtain a cubic phase with the structural characteristics to successfully reconstitute the large membrane protein into its lipidic bilayer.

**A robust toolbox for large membrane proteins crystallization.** In order to assess whether the swollen mesophases are a suitable system for membrane protein crystallization, a few obstacles have to be cleared first. To start, when using electrostatically swollen mesophases, we need to pay attention to the addition of crystallizing salts, such as sodium chloride in concentrations higher than 100–150 mM, which can annihilate the electrostatic swelling effects. This limitation was overcome via a systematic pre-screening of protein buffers and detergents, which led to identifying the ideal protein buffers suitable for our scope. More specifically, after expression and purification of all proteins considered, the protein buffer was exchanged to a low-salt buffer allowing both stable solutions and in meso crystallization without disrupting the swollen mesophases.

Secondly, to demonstrate feasibility of crystallization from the swollen mesophases, a control experiment with a model membrane protein, which is crystallisable in meso also via conventional lipidic mesophases, is needed. To this end, we selected the highly stable transmembrane domain of the *Escherichia coli* virulence factor, intimin (PDB: 5G26)[29], previously crystallized in meso from an MP-based cubic phase[15,16], and purified into a low-salt/low-detergent buffer (see Materials and Methods) preserving the swelling of the MP/DSPG/water systems. We then subjected intimin to crystallization trials using the 5 and 10 wt% DSPG systems yielding at 80% hydration swollen Pn3m and mixed Ia3d/$L_\alpha$ phases, respectively and immediately observed protein crystal growth from both systems confirmed by means of UV-microscopy (Supplementary Figure 2). Moreover, the crystals grown from the swollen systems were similar in shape and size to those previously obtained from MP-based cubic phases[15], confirming the suitability of the swollen mesophases for membrane protein crystallization. This is in agreement with the observations of Sparr et al.[26] who successfully crystallized bacteriorhodopsin from a swollen MO-based mesophase, observing both improvement of crystal quality and growth speed compared to the crystals of the same protein obtained in standard MO-based mesophases.

We then moved on to assess the full potential of the swollen mesophases in crystallizing membrane proteins with large ECDs

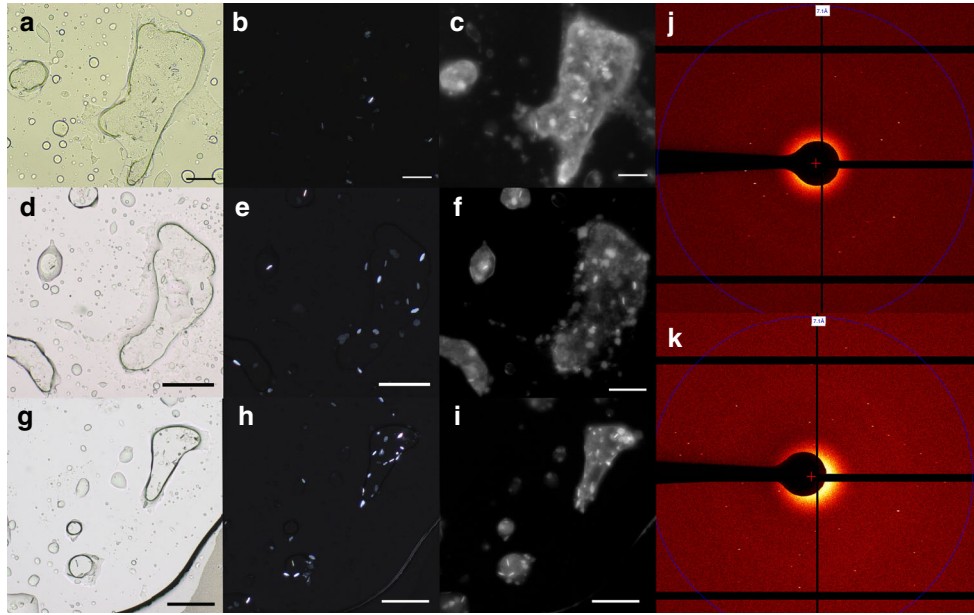

**Fig. 4** GLIC protein crystals grown in meso using DSPG–MP. GLIC protein crystals were grown in meso at 20 °C using 5 wt% DSPG/MP (**a–c**), 8 wt% DSPG/MP (**d–f**), and 10 wt% DSPG/MP (**g–i**) swollen mesophase systems (2:8 lipid to protein ratio), with crystal diffraction patterns (**j**) and (**k**) from the SLS crystallography beamline for crystal grown using 10 wt% DSPG/MP mesophase system. Crystallization conditions were 0.2 M $(NH_4)SO_4$, 0.02 M NaCl, 0.02 M Na Act 4 pH, 33% v/v PEG200. GLIC protein crystals are shown here under brightfield (**a**, **d**, **g**), cross-polarized (**b**, **e**, **h**), and UV fluorescence (**c**, **f**, **i**) microscope at day 7. Scale bar is 100 μm on panels **a–c** and 200 μm on panel **d–i**

by selecting a membrane protein that would be inaccessible to the "classical" in meso approach due to the prohibitive size of its hydrophilic domain (Fig. 3b). The GLIC is a pentameric 174 kDa (1585 amino acid) membrane protein, characterized by a large extracellular domain that surpasses in size the water channel diameter of most LCPs. GLIC was thus purified in a low-salt/low-detergent buffer that would not disrupt the swollen mesophase during protein reconstitution. Crystallization trials were then setup using three distinct lipidic systems containing 5, 8 and 10 wt% DSPG, yielding highly swollen Pn3m, Ia3d, and Ia3d/$L_\alpha$ phases prior to the addition of the crystallization buffer. Protein crystal growth was then consistently observed in all the tested systems (Fig. 4a, d, g) and confirmed by means of cross-polarized microscopy (Fig. 4b, e, h), UV microscopy (Fig. 4c, f, i), and single-crystal diffraction (Fig. 4j, k), confirming the successful crystallization of GLIC using the in meso approach. Importantly, control experiments run with the same crystallization buffers from MP-based non-swollen mesophases produced no visible or diffracting crystals.

Interestingly, although the morphologies of the crystals were identical for the three tested systems (i.e., rod-like crystals with a maximum length of ~30 μm), the time required for crystal formation varied based on the initial symmetry of the hosting mesophase. Respectively, the crystallization from an initial double-gyroid symmetry (8 and 10 wt% DSPG) yielded crystals after ~7 days, whereas crystallization from an initial double-diamond symmetry required slightly longer time (10 days) for crystallization to take place. This may be related to different protein diffusion rates in the lipid bilayer of the different cubic mesophases[16].

Single-crystal diffraction experiments from the in meso protein crystals allowed us to resolve the structure of the pentameric membrane protein (Fig. 5, Table 1) validating the use of the swollen cubic phases for membrane protein crystallization, thus opening previously unexplored pathways in protein structural biology. The obtained resolution of 6 Å is modest, but typical for first hits of membrane protein crystallization trials[30]. Further

optimization of crystallization conditions would presumably enable higher resolution to be reached; however, this was not the purpose of the present study. More importantly, in-depth structural analysis revealed that the in meso grown GLIC crystals crystallized in a completely different and unreported space group for this protein, exhibiting a tighter packing arrangement, with less solvent content (56.3% solvent content) compared to loose crystals grown via vapor diffusion techniques (typically 76.7% solvent content)[30]. Moreover, the structure obtained from in meso grown crystals shows the GLIC molecules to be in a closed state (see Structure solution and refinement in the Methods section and Supplementary Figure 3), in spite of the presence of $H_3O^+$ ions at the crystallization pH of 4, which normally stabilize the open state of the channel, as found in the vast majority of reported GLIC structures[30]. We conclude that this stabilizing effect, which allows the "entrapment" of this observed protein conformation, might be due to the specific protein–protein contacts existing in the newly generated crystal packing, which further exemplify the benefits associated with the swollen in meso crystallization method for membrane proteins with large ECDs.

## Discussion
In conclusion, mixing charged phospholipids within a host MP lipid bilayer in the presence of water results in the formation of highly swollen, thermodynamically stable cubic phases with interesting structural features; the most notable being the re-entrant double-gyroid and double-diamond bicontinuous cubic phases upon increasing hydration levels. Although the physical mechanisms behind these changes are not yet fully understood, the observed phase behavior suggests competing effects at play in the establishment of the observed mesophases.

The ultra-swollen mesophases were successfully used for the crystallization of a membrane protein, GLIC, with large ECDs otherwise inaccessible to the conventional in meso crystallization method. Analysis of the ensued crystals resulted in the structure of a large membrane protein, obtained from single-crystal diffraction experiments grown using the swollen in meso

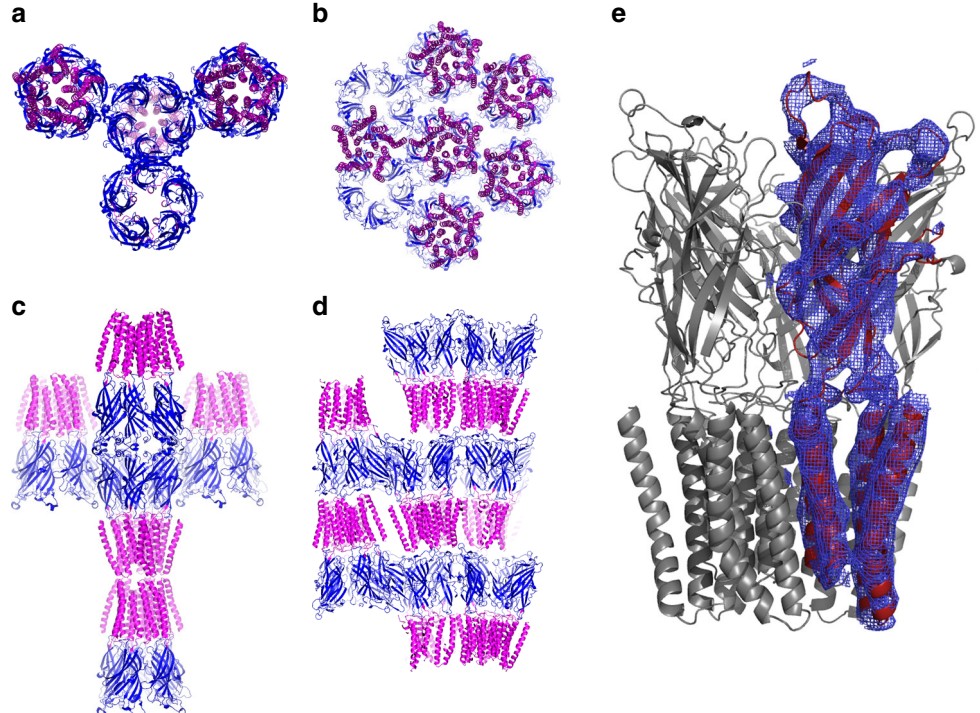

**Fig. 5** GLIC protein structure obtained from in meso grown crystals. **a–d** Crystal packing of the GLIC protein in crystals grown from vapor diffusion, in the C2 space group (**a**, **c**), compared with the packing obtained by in meso crystallization, in space group C222₁ (**b**, **d**). **e** Structure of the GLIC protein (PDB ID 6F7A) from the in meso grown crystals. The electron density is contoured at 1σ, and shown representatively only around one of the units of the pentamer for clarity

**Table 1 Data collection and refinement statistics for the in meso GLIC structure**

|  | GLIC |
|---|---|
| **Data collection** |  |
| Space group | C222₁ |
| **Cell dimensions** |  |
| *a, b, c* (Å) | 75.94, 208.22, 255. |
| *a, b, c* (°) | 90, 90, 90 |
| Resolution (Å) | 48.2–6.001 (6.215–6.001)ᵃ |
| $R_{meas}$ | 0.419 (3.25) |
| *I* / *sI* | 4.99 (0.94) |
| Completeness (%) | 98 (99) |
| Redundancy | 5.4 |
| **Refinement** |  |
| Resolution (Å) | 48.2–6.00 |
| No. of reflections | 5287 (533) |
| $R_{work}$/$R_{free}$ | 0.2861/0.3187 |
| **No. of atoms** |  |
| Protein | 11,682 |
| Ligand/ion | 0 |
| Water | 0 |
| **B-factors** |  |
| Protein | 307.36 |
| **R.m.s. deviations** |  |
| Bond lengths (Å) | 0.007 |
| Bond angles (°) | 0.97 |

Values in parentheses are for highest-resolution shell
ᵃTen wedges of 15° from different microcrystals were merged (see Methods section)

crystallization method. The crystals were found to be organized in a different space group compared to crystals of the same protein grown by vapor diffusion, with the noteworthy additional features of tighter packing of the protein and the stabilization of its closed state as opposed to the more conventional open state found in loose crystals obtained by vapor diffusion.

These results showcase the possibility of expanding the reach of the in meso crystallization method to previously unreachable proteins and to design protein crystals into different space groups, packing efficiency, and activation states inaccessible via other crystallization methods, providing powerful tools of significance to membrane protein structural biology.

## Methods

**Materials**. MP (M-219) was purchased from Nu-Chek Prep (Minnesota, USA). 1,2-Distearoyl-sn-glycero-3-phospho-rac-glycerol, sodium salt (DSPG, 560400), was kindly provided by LIPOID PG (Steinhausen, Switzerland). LCP glass plates with a 100 μm double-sided tape spacer and 200 μm plastic seals were purchased from Molecular Dimensions. All the salts and detergents needed for the protein purification and preparation of the crystallization buffers were purchased from Sigma Aldrich, unless otherwise stated.

**Mesophase sample preparation**. A range of MP lipidic mixtures with 3, 5, 6, 7, 8, 9, 10, 11, and 12 wt% DSPG were initially prepared to determine the specific phospholipid/MP ratios to swell the pn3m (i.e., 5 wt% DSPG at 80% hydration) and ia3d (i.e., 8 wt% DSPG at 80% hydration) lyotropic cubic mesophase (Supplementary Figure 1). Lipidic mixtures were prepared by co-dissolving the appropriate weighed amounts of dry lipids, MP and DSPG, in chloroform. Solvent was completely removed by rotary evaporation. Mesophase samples were then prepared by mixing weighed quantities of DSPG/MP lipid and Milli-Q water (i.e., 40–80% hydration) inside sealed pyrex tubes by vortexing at room temperature until a homogenous mixture was obtained. The prepared mesophase was then allowed to equilibrate at room temperature for 72 h.

**Small-angle X-ray scattering**. Data were collected at the SAXS/WAXS beamline at the Australian Synchrotron. Data were obtained at a constant temperature of 20 °C. The experiments used a micro-sized beam of dimensions 100 μm × 100 μm, using a wavelength λ = 1.0322 Å (12.0 keV) for the MP-based samples, with a typical flux of $1.2 \times 10^{13}$ photons per second and a 1 s exposure time. Previous work suggests that radiation dosages in the range used in this study are unlikely to affect the mesophase significantly[31] 2D diffraction images were recorded on a Pilatus 1M detector, which offers very low noise, a large dynamic range and rapid

data collection over a large active area. Dead space due to intermodule gaps is overcome by radial integration with the detector slightly offset to ensure complete data coverage. The obtained diffraction images were integrated into 1D diffraction spectra using the ScatterBrain IDL software developed in house by the research team of the Australian Synchrotron. The obtained 1D spectra were then analysed using origin, for both peak assignment and calculations of the phase structural parameters.

SAXS measurements were also performed on a Bruker AXS Micro, with a microfocused X-ray source, operating at voltage and filament current of 50 kV and 1000 μA, respectively. The Cu Kα radiation ($\lambda Cu\ K\alpha = 1.5418$ Å) was collimated by a 2D Kratky collimator, and the data were collected on a 2D Pilatus 100K detector. The scattering vector $Q = (4\pi/\lambda)\sin\theta$, with $2\theta$ being the scattering angle, was calibrated using silver behenate. Data were collected and azimuthally averaged using the Saxsgui software to yield 1D intensity vs. scattering vector $Q$, with a $Q$ range from 0.004 to 0.5 Å$^{-1}$. For all measurements the samples were placed inside a stainless steel cell between two thin replaceable mica sheets and sealed by an O-ring, with a sample volume of 10 μL and a thickness of ~1 mm. Measurements were performed at 20 °C, and samples were equilibrated for 15 min before measurements, whereas scattered intensity was collected over 20 min.

In addition, SAXS measurements were also performed at the X06DA PXIII Source beamline at the Swiss Light Source, Paul Scherrer Institute (Villigen, Switzerland), equipped with a Pilatus 2M detector (Dectris, Baden-Dättwil, Switzerland). The photon energy was set to 5.975 keV, the in-air sample-to-detector distance was 800 mm and the sample-to-beamstop distance was 65 mm. The beam size was 50 × 90 μm$^2$, with a flux of $3.5 \times 10^{11}$ photons per second and exposure time of 10 s. Samples were loaded in quartz capillaries and measured at room temperature.

**Expression and purification of intimin.** Detailed information on the construction of the plasmids containing the intimin *E. coli* O157:H7 gene has been previously described[29]. The construct was kindly provided by Dr. Susan K. Buchanan from the National Institute of Diabetes and Digestive and Kidney Diseases, Bethesda, MD 20892, USA and used as received without any further modifications.

Briefly, the vector containing the *E. coli* O157:H7 gene was transformed into BL21 (DE3) cells (Novagen—Merck Millipore, Darmstadt, Germany) which were grown in TB media (50 μg mL$^{-1}$ kanamycin) at 20 °C for 2–3 days while shaking at 220 rpm until they reached a terminal OD of 15–20. The cells lysed using a probe sonicator (Misonix S4000), in 30 s bursts at 60% amplitude. Membranes containing the desired protein were harvested by ultra-centrifugation (160,000×*g*, 60 min, 4 °C). Membrane proteins were solubilized by resuspension in solubilization buffer (50 mM Tris pH 8.0, 200 mM NaCl, 20 mM Imidazole, 5% Elugent (Calbiochem)) and left stirring O/N at 4 °C. The next morning, the sample underwent ultra-centrifugation (250,000×*g*, 60 min, 4 °C) to remove insoluble material. The protein was purified using a combination of affinity and ion-exchange chromatography. Fractions containing protein were then buffer exchanged in low-salt buffer (20 mM Tris pH 8.0, 25 mM NaCl, 2% OG) and concentrated in a YM30 Amicon Ultra concentrator (Millipore) to prepare for crystallization experiments.

**Expression and purification of GLIC.** GLIC was expressed from a pET20 plasmid, kindly provided by Marc Delarue (Pasteur Institute, CNRS URA 2185, Paris, France), as a fusion protein with maltose-binding protein (MBP), essentially as described previously[30]. The expression coding sequence contained an N-terminal signal peptide, followed by MBP and a thrombin cleavage site, preceding GLIC. Briefly, BL21(DE3) cells (Novagen—Merck Millipore, Darmstadt, Germany), harboring the pET20-MBP-thrombin-GLIC plasmid were grown at 37 °C in terrific broth containing 100 μg mL$^{-1}$ ampicillin, to an optical density at 600 nm of 1.6. The culture was induced with 0.1 mM IPTG and growth continued for a further 16 h at 20 °C. Cells were harvested by centrifugation and lysed by three passes through an Emulsiflex C5 homogenizer (Avestin, Ottawa, Canada) at 15,000 psi in buffer A (20 mM Tris pH 7.6, 300 mM NaCl, with "complete protease inhibitor cocktail (Roche)". Lysate was clarified by centrifugation (7000×*g*, 20 min, 4 °C) and membranes pelleted from the supernatant by ultra-centrifugation (100,000×*g*, 60 min, 4° C). Membrane proteins were solubilized from the membrane pellet in buffer A with 2% *n*-Dodecyl-β-D-Maltopyranoside (DDM, Anatrace, Maumee, OH, USA) at 4 °C overnight, then clarified by ultra-centrifugation (100,000×*g*, 60 min, 4 °C). GLIC was purified from the supernatant by amylose affinity chromatography, with elution in 20 mM Maltose and pooled fractions further purified by size-exclusion chromatography (Superdex-200 10/300 GL, GE Life Sciences) in buffer A with 0.02% DDM. Fractions corresponding to MBP-GLIC pentamer were concentrated and digested with thrombin (Merck, KGaA, Darmstadt, Germany) at room temperature overnight. MBP and thrombin were removed by a further round of size-exclusion chromatography and the GLIC pentamer concentrated to 10.0 mg mL$^{-1}$ and exchanged into 50 mM NaCl, 20 mM Tris pH 7.6, using an Amicon Ultra-2 mL 30 K concentrator (Millipore), to prepare for crystallization.

**Crystallization.** The protein solution mixed in a 50:50 (volume:volume) ratio with molten MP or in a 80:20 (volume:volume) ratio with either of the MP:DSPG mixtures before being dispensed in 200 nL aliquots onto the surface of a LCP glass plate with double-sided tape spacer. One microliter of crystallant was dispensed on top, and the experiment sealed with a 200 μm plastic seal. All dispensing was

performed with a Mosquito LCP machine equipped with a humidity chamber (TTPLabtech, UK). The completed plates were incubated at 20 °C and imaged in a Minstrel HT/UV imaging system (Rigaku).

GLIC protein was syringed mixed manually with MP (1:1 lipid-to-protein ratio) for our control plates, whilst experimental plates were setup using GLIC protein with DSPG/MP (at 5, 8, and 10 wt%) using a 1:4 lipid-to-protein ratio, to enable 80% w/w hydration of the mesophase. Once homogenously mixed, dispensing was automated using robotic nano-drop liquid handler, Crystal Phoenix (Art Robbin Instruments, California, USA), for dispensing 50 nL mesophase to 800 nL of precipitant in per well. Samples were prepared in 96-well SBS LCP plates, glass cover, and 60 μm spacer (Swissci, Zug, Switzerland). Twenty plates were set up using five different commercially available screens: MemStart (MD1-21), MemSys, (MD1-25), ShotGun1, and MemGold (Molecular Dimensions, Newmarket, UK), and HR2-453 Screen (Hampton Research, California, USA), were used in screening for crystallization conditions in the normal (control plates) and swollen systems (experimental plates). Rock Maker (Ver. 3.10.0.103, Formulatrix, Massachusetts, USA) was used to collect comprehensive images (i.e., condenser, cross-polarized, and UV images) of the wells at 20 °C for analysis.

GLIC crystals were grown in meso from well F11 (0.2 M (NH$_4$)$_2$SO$_4$, 0.02 M NaCl, 0.02 M Na Act 4 pH, 33% v/v PEG200) of the Mem Gold Screen (Molecular Dimensions, Newmarket, UK) in all DSPG/MP lipidic mixtures (5, 8, and 10 wt%) prepared. PEG200 was selected since low molecular weight PEG (<200 g/mol), is known to preserve the symmetry of cubic phases, including those based on MP[32]. All these wells, containing GLIC protein crystals, were fished using mesh cryo-loops and stored in liquid nitrogen until beamtime without any cryo-protectants. The crystal diffraction data obtained in this study come from the only microcrystals that were successfully fished from a well prepared using 10%DSPG/ MP (Fig. 4).

**Diffraction data collection and processing.** The data were collected at the X06SA-PXI beamline at the Swiss Light Source (Villigen, Switzerland), using a 10 × 10 μm$^2$ beam with photon energy of 12.39 keV, at the full flux of $3\times10^{11}$ photons per second. Images were recorded with an EIGER 16M detector (Dectris, Switzerland) placed at 400 mm distance with exposure time of 0.1 s for a rotation of 0.1° per frame. Crystals of 10–30 μm in size diffracting to low resolution (about 6 Å) were found on the mesh mounts using a systematic rastering procedure[32] followed by in-house developed automatic data collection for microcrystals (CY+ protocol) over a total range of 15° on each crystal. The data completeness was maximized by collecting new ranges of 15° from a different orientation on the best diffracting crystals, until radiation damage caused a significant decrease of the diffraction signal. The data were processed with XDS[33], scaled and merged with XSCALE[34], using an in-house script provided by Shibom Basu. A complete dataset was obtained by merging the best ten wedges of 15°, however the nominal resolution was of only 6.0 Å (using a resolution cutoff of $I/\sigma(I) = 1$). The space group, confirmed by POINTLESS[35], was C222$_1$ with unit cell parameters $a = 75.94$Å, $b = 208.22$Å, $c = 255.29$Å, $\alpha = \beta = \gamma = 90°$. Complete data collection statistics are presented in Table 1 and Supplementary Table 1.

**Structure solution and refinement.** The structure was solved by molecular replacement (MR) and refined with the Phenix suite[36]. The PDB entries 4HFI (open channel state) and 4NPQ (resting/locally-closed state) were used as search model for MR and reference model for constraints in the low-resolution structure refinement. The other options used in *phenix.refine* for refinement of the low-resolution structure were rigid-body refinement in the first round, NCS constraints, group B-factor refinement, and secondary structure constraints. As the resolution achieved was low, mainly secondary structure elements were identifiably visible in the electron density map, such as α-helixes in the transmembrane domain, β-sheets, and loops in the extracellular domain (Fig. 5). When initially using the open channel state as MR and restraints model, characteristic residual difference densities in the $F_o−F_c$ fourier difference map were observed in the inner ring of transmembrane α-helices upper half, near the extracellular domain (Supplementary Figure 3), which clearly pointed to a predominantly closed form of the channel[30]. The difference densities were not present when the locally-closed state was used as MR and restraints model, and the refinement $R_{work}$ and $R_{free}$ values decreased significantly compared to the open state case, reflecting the improved agreement of the model and data. Final values of $R_{work}/R_{free}$ were 0.28/0.32. The Ramachandran statistics were 94% favored, 5.3% allowed and 0.32% outliers, and there were 1.2% rotamer outliers. Complete refinement statistics are presented in Table 1 and Supplementary Table 1. The solvent content was analyzed using the program RWCONTENTS in the CCP4 suite[35]. To check the absence of significant model bias, a composite omit map was created using the software suite Phenix (Supplementary Figure 3). Figures were produced using the software PyMOL (The PyMOL Molecular Graphics System, Version 1.1, DeLano Scientific LLC).

**Evaluation of structural parameters.** To determine the evolution of the structural parameters with hydration level, i.e., the size of the water channels, SAXS data information on the lattice were combined with the composition of the samples. To calculate the diameter of the water channel for the three bicontinuous cubic phases (Ia3d, Pn3m, and Im3m), triply periodic minimal surfaces arguments were used

and the following equation from Tuner et al.[37] was applied:

$$\phi = 2A_0\left(\frac{l}{a}\right) + \frac{4}{3}\pi x\left(\frac{l}{a}\right)^3, \qquad (1)$$

where $a$ is the lattice parameter as measured by SAXS, $\varphi$ is the lipid volume fraction, which can be obtained knowing the water content and the density of MP ($\rho = 0.982\ g\ cm^{-3}$), $l$ is the length of the lipid chains, $A_0$ and $\chi$ are respectively the ratio of the area of the minimal surface in a unit cell to (unit cell volume)$^{2/3}$ and the Euler–Poincaire characteristic, which have the following values depending on the specific cubic phase: $A_0 = 3.091$ and $\chi = -8$ for Ia3d; $A_0 = 1.919$ and $\chi = -2$ for Pn3m; $A_0 = 2.345$ and $\chi = -4$ for Im3m. Following Briggs et al.[38], we derive the radius of the water channels by:

$$\overline{XX}(\text{Ia3d})\ r = 0.248a - l \qquad (2a)$$

$$(\text{Pn3m})\ r = 0.391a - l \qquad (2b)$$

$$(\text{Im3m})\ r = 0.305\ a - l \qquad (2c)$$

**Data availability**. Data supporting the findings of this manuscript are available from the corresponding author upon reasonable request. The GLIC structure is deposited in the PDB under the accession code 6F7A.

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

## Acknowledgements

We acknowledge the use of the SAXS/WAXS beamline at the Australian Synchrotron and the C3 Collaborative Crystallization Center, CSIRO, Parkville, Australia. We would like to acknowledge the use of the X06SA-PXI beamline at the Swiss Light Source, Paul Scherrer Institute, Villigen, Switzerland. We thank Shibom Basu for providing the automatic data analysis script. We also thank Beat Blattmann and Céline Stutz-Ducommun from the Protein Crystallization Center (Department of Biochemistry, University of Zürich) for their continuing and expert support. We also thank Lipoid AG, Steinhausen, Switzerland for providing DSPG phospholipid for our experiments. We also thank Dr. Salvatore Assenza for his kind help with program Mathematica (Wolfram Research, Inc., Mathematica, Version 11.1, Champaign, IL (2017) for 3D models. Support from the Swiss National Science Foundation Sinergia Grant CRSII2_154451 and Australian NHMRC grant APP1104259 are gratefully acknowledged.

## Author contributions

A.Z. prepared, collected and analysed SAXS samples, expressed, purified and crystallized protein samples, prepared the figures, and wrote the manuscript. J.Y.T.C. prepared, collected and analysed SAXS samples, crystallized/harvested protein crystal for analysis, prepared the figures, and wrote the manuscript. I.M. collected synchrotron data, solved and refined protein structure, and wrote the manuscript. L.S. and B.A.C. expressed, purified, and analysed protein samples. C.S. collected SAXS data. C.J.D. contributed to supervision of part of the experimental work, experimental results interpretation, and wrote the manuscript. R.M. designed and directed the study, run synchrotron

experiments, analysed and interpreted the data, and wrote the manuscript. All authors gave final approval to the manuscript.

## Additional information

**Competing interests:** The authors declare no competing financial interests.

