## [Peer Review File · Nature Communications]

Reviewers' comments:

Reviewer #1 (Remarks to the Author):

This is an interesting manuscript, which is likely to be of considerable interest to a wide range of readers. It describes some progress towards extending the in-cubo crystallization technique to membrane proteins having large extracellular domains (ECD). Although a number of other groups have reported the swelling of inverse bicontinuous cubic phases to large lattice parameters (and hence large water channels) by addition of small amounts of charged lipids to a monoglyceride, there are a number of novel features reported here:

- a) The observation of re-entrant highly swollen Ia3d and Pn3m cubic phases at high water contents;
- b) The use of the highly swollen cubic phases to crystallize a membrane protein with a large ECD, and solve its structure (although to relatively low resolution).

One minor specific point: In Figure 1, Pn3m is incorrectly denoted 'PN3M' (twice).

Reviewer #2 (Remarks to the Author):

The Authors expand the application of in-meso crystallization method by using ultra-swollen bicontinuous cubic phase with 5times larger channel sizes than those of the commonly used. This allows them to crystallize larger proteins which is of special interest and a "hot topic" for structural biology as well as for the mechanistic studies of integral proteins in biomimetic environment. They show the strategy on the example of in-meso crystallization of a channel protein GLIC. The Authors should focus more in their discussion on a similar approach that was described for the crystallization of bR by Sparr, Engstrom et al. (the protein was smaller but the authors also observed improvement of the quality of the protein crystals by widening the channel widths to accommodate better the protein)

1. The same series of SAXS for MP/H₂O without DSPG under different hydration conditions should be shown for comparison (Fig. 2)
2. The signals e.g. are hardly seen of Fig 2b. On a log scale they could be perhaps better developed (e.g. for 75% of hydration).
3. Information on the method of channel width evaluation should be included. What is the value of lipid length, l used in this calculation.
4. Is it valid to assume that doping does not change the structure of the mesophase? It should be added that for crystallization 33% PEG200 is added (as mentioned in the caption of Fig 4. Does the PEG200 presence influence phase symmetry?
5. What are the differences in the protein crystals when phases of Ia3d, Im3m or Pn3m are used for crystallization? Comment should be added on the phase symmetry effect on the crystal structure or size.

Reviewer #3 (Remarks to the Author):

The authors present a new lipid mixture for in-meso crystallization of membrane proteins with large extracellular domains. This would also be of interest for those working on membrane proteins in complex with large cytosolic regulatory or signaling proteins. While this new mixture is promising, it remains to be fully validated. The authors obtained crystals of *Gloeobacter* Ligand-gated Ion Channel, which has a very large extracellular domain. However, the resolution was only

6Å. While this confirms the ability of the mixture to accommodate a large extramembrane domain, the resolution is not sufficient to provide meaningful structural insights. As such, I don't believe it merits publication in Nature Communications.

Other concerns:

The lipid mixture appears to be incompatible with the additions of cholesterol. "Cholesterol was avoided to allow the use of the ensued mesophases at sufficiently low temperature for in-meso crystallization." Is it the case that no amount of cholesterol would be tolerated. This should be more fully addressed as many membrane proteins are dependent on cholesterol for stability and function.

The authors should provide more of a discussion of 7.7 MAG described by Caffrey's group. This lipid also provides a larger aqueous pore and can be used in combination with cholesterol.

Answer to Referees:

Reviewers' comments:

Reviewer #1 (Remarks to the Author):

“This is an interesting manuscript, which is likely to be of considerable interest to a wide range of readers. It describes some progress towards extending the in-cubo crystallization technique to membrane proteins having large extracellular domains (ECD). Although a number of other groups have reported the swelling of inverse bicontinuous cubic phases to large lattice parameters (and hence large water channels) by addition of small amounts of charged lipids to a monoglyceride, there are a number of novel features reported here:

a) The observation of re-entrant highly swollen Ia3d and Pn3m cubic phases at high water contents;

b) The use of the highly swollen cubic phases to crystallize a membrane protein with a large ECD, and solve its structure (although to relatively low resolution).”

We would like to thank the referee for his/her assessment of the manuscript and the novel features reported within.

“One minor specific point: In Figure 1, Pn3m is incorrectly denoted 'PN3M' (twice).”

We acknowledge the referee’s comment and have modified Figure 1 accordingly.

Reviewer #2 (Remarks to the Author):

“The Authors expand the application of in-meso crystallization method by using ultra-swollen bicontinuous cubic phase with 5times larger channel sizes than those of the commonly used. This allows them to crystallize larger proteins which is of special interest and a “hot topic” for structural biology as well as for the mechanistic studies of integral proteins in biomimetic environment. They show the strategy on the example of in-meso crystallization of a channel protein GLIC.”

We thank the referee for his/her overview of the manuscript relevance.

“The Authors should focus more in their discussion on a similar approach that was described for the crystallization of bR by Sparr, Engstrom et al. (the protein was smaller but the authors also observed improvement of the quality of the protein crystals by widening the channel widths to accommodate better the protein)”

We acknowledge the referee’s comments and we note that the article mentioned by the referee is already cited in the manuscript as reference 26. We nonetheless have expanded the discussion and added the following paragraph into the discussion part of the manuscript: “This is in agreement with the observations of Sparr et al.,²⁶ who successfully crystallized bacteriorhodopsin from a swollen monoolein-based mesophase, observing both improvement of crystal quality and growth speed compared to the crystals of the same protein obtained in standard monoolein-based mesophases.”

“1. The same series of SAXS for MP/H2O without DSPG under different hydration conditions should be shown for comparison (Fig. 2)”

The monopalmitolein:water phase diagram is known and has been previously investigated and discussed at length in the following cited manuscript: “Briggs, J. The Phase Behavior of Hydrated Monoacylglycerols and the Design of an X-Ray Compatible Scanning Calorimeter. PhD Thesis, The Ohio State University (1994).” – reference number 28. This document is also available online free of charge at:

https://etd.ohiolink.edu/rws_etd/document/get/osu1224183084/inline.

In order to avoid redundancy we did not to plot the monopalmitolein:water mesophases at different hydration levels within Figure 2; however we give the MP/H2O reference mesophase structure in scale at Figure 1 and compare it with the corresponding swollen one by addition of DSPG.

“2. The signals e.g. are hardly seen of Fig 2b. On a log scale they could be perhaps better developed (e.g. for 75% of hydration).”

The 2D SAXS spectra in Figure 2b show the scattered intensities already on a logarithmic scale plotted versus the scattering vector q (the latter in a linear scale). The reason why some spectra are less visible lies with the fact that the 2D spectra have been normalized and stacked within a 3D plot that enhances the visual impact of the results. 2D plots of the

MP:DSPG:water system in log-lin scale have been added in the S.I. where the signal intensity can be better observed.

“3. Information on the method of channel width evaluation should be included. What is the value of lipid length, l used in this calculation.”

The information on the mathematical model used to calculate the structural parameters of observed mesophases was already added in the supporting information. The length l is obtained by solving supporting eq.1 for l with all the other parameters known, and then the radii of water channels are obtained by applying eq. 2a to 2c.

To specify this point, we have added the following text in the main manuscript: “(for the calculation of the channel radii see supporting information and supporting equations 1 to 2a-2c).”

“4. Is it valid to assume that doping does not change the structure of the mesophase? It should be added that for crystallization 33% PEG200 is added (as mentioned in the caption of Fig 4. Does the PEG200 presence influence phase symmetry?”

The referee asks here an interesting question. In general, addition of crystallization buffers with high molecular weight PEG on top of the protein-containing mesophases may influence the structure of the starting mesophase. For example, it has been previously observed that addition of a crystallization buffer containing a PEG400 can push the starting cubic phase towards a less structured/more fluidic sponge phase within a matter of hours.¹⁵ Another more comprehensive and systematic study on the topic, however, -see van t’Hag et al. *Cryst. Growth Des.* 2014, 14, 1771–1781- shows that crystallization buffers containing low molecular weight PEG (<400) do preserve the cubic symmetry of monopalmitolein-based cubic phases. Since it is now known that a gradient of symmetry exists from the crystallization buffer to the inner mesophase,¹⁵ exact details may only be accessible via a microfocused x-ray synchrotron scanning beam. More importantly, as was previously discussed,¹⁵ the final structural symmetry of the hosting lipidic phase does not play a major role on the crystallization process since the necessary order-to-order transitions that are imperative for crystal growth take place regardless of the final state of the system.

The following sentence has been added to the manuscript in the method section, including additional reference 32:

“PEG200 was selected since low molecular weight PEG (<200g/mol), is known to preserve the symmetry of cubic phases, including those based on monopalmitolein.³²”

Ref 32: van t’Hag et al. *Cryst. Growth Des.* 2014, 14, 1771–1781

5. What are the differences in the protein crystals when phases of Ia3d, Im3m or Pn3m are used for crystallization? Comment should be added on the phase symmetry effect on the crystal structure or size.

The crystals differ in size and growth kinetics when generated from different initial symmetries, but they do preserve the same crystal habit, from which it can be inferred that the space group is the same. Thus, the main difference is in growth size and kinetics. This has been discussed in the manuscript via the following text:

“Interestingly, although the morphologies of the crystals were identical for the three tested systems (i.e. rod-like crystals with a maximum length of approximately 30 μm), the time required for crystal formation varied based on the initial symmetry of the hosting mesophase. Respectively, the crystallization from an initial double gyroid symmetry (8wt% and 10wt% DSPG) yielded crystals after approximately 7 days, whereas crystallization from an initial double diamond symmetry required slightly longer time (10 days) for crystallization to take place. This may be related to different protein diffusion rates in the lipid bilayer of the different cubic mesophases.¹⁶”

Parenthetically, we note that the preserved symmetry of the crystal can only be inferred from the crystal habits, and not the direct crystal diffraction because crystals grown from Pn3m (Im3m is not possible as not coexisting with excess water) were smaller than those obtained by Ia3d and thus could not be successfully fished and shoot on the x-ray beam.

Reviewer #3 (Remarks to the Author):

“The authors present a new lipid mixture for in-meso crystallization of membrane proteins with large extracellular domains. This would also be of interest for those working on membrane proteins in complex with large cytosolic regulatory or signaling proteins.”

We are in full agreement with the referee on the potential interest that a functioning swollen lipidic system can generate in the membrane structural biology field.

“While this new mixture is promising, it remains to be fully validated. The authors obtained crystals of Gloeobacter Ligand-gated Ion Channel, which has a very large extracellular domain. However, the resolution was only 6Å. While this confirms the ability of the mixture to accommodate a large extramembrane domain, the resolution is not sufficient to be provide meaningful structural insights. As such, I don’t believe it merits publication in Nature Communications.”

We are in perfect agreement with the referee that resolution of the GLIC crystal at 6Å is low as already discussed in the original manuscript: “*The obtained resolution of 6Å is modest, but typical for first hits of membrane protein crystallisation trials.*³⁰ *Further optimisation of crystallisation conditions would presumably enable higher resolution to be reached, however this was not the purpose of the present study.*”; However, resolution is of secondary importance in this work and we believe this does not at all minimize the significance of our findings. In the structural biology pipeline, obtaining initial crystallisation hits and optimizing the resolution of the crystals are two different steps. Our goal here was to address the

crystallization bottleneck by developing a novel swollen lipidic cubic phase allowing reconstitution and crystallization of large membrane proteins otherwise inaccessible to the “classic” in-meso approach. The GLIC protein was therefore chosen as a representative model system with a large extracellular domain, precisely because its structure is already known, which makes comparison with previous work more meaningful. The crystallization hits yielded a resolution of 6Å, which is a typical resolution for first hits in such challenging membrane protein cases. The optimization step required to bring the quality of a first crystallization hit of a membrane protein, like our 6Å dataset, to a structurally meaningful resolution is challenging in itself and can take substantial effort, which was not justified in the scope of this work. For these reasons, a resolution better than 6Å was not presently pursued. However, carrying typical optimization work using LCP-suitable screens will be absolutely possible for biology researchers using this lipid system in the future.

In spite of the modest resolution of the data from the obtained crystals, a full dataset was collected, and its quality was sufficient for solving the protein structure and to come to the main conclusions of our work. The secondary structure of the membrane protein (in particular the alpha helices in the transmembrane domain) is clearly recognizable, to the level of detail expected for a structure at 6Å resolution. This univocally demonstrates that the crystallization of the targeted protein in the lipid system was successful, which is the main point of the present work. Moreover, it turns out that this structure actually does contain useful structural information in spite of its modest level of detail, as outlined in our manuscript. The space group obtained has never been reported before for the GLIC protein, as demonstrated by the successful deposition of this structure in Protein Data Bank (PDB ID 6F7A - presently release status is: HOLD FOR PUBLICATION). Finally, the crystal contacts and low solvent content appear particularly interesting, as well as the stabilisation of a specific conformation in unusual conditions (closed vs open). All these elements demonstrate that the resolution at 6Å is of secondary importance in this work.

“Other concerns:

The lipid mixture appears to be incompatible with the additions of cholesterol. “Cholesterol was avoided to allow the use of the ensued mesophases at sufficiently low temperature for in-meso crystallization.” Is it the case that no amount of cholesterol would be tolerated. This should be more fully addressed as many membrane proteins are dependent on cholesterol for stability and function.

The authors should provide more of a discussion of 7.7 MAG described by Caffrey’s group. This lipid also provides a larger aqueous pore and can be used in combination with cholesterol.”

The referee is misled by a sentence poorly formulated in our previous submission. Indeed, our host lipid monopalmitolein does support –down to room temperature and to a very large concentration- the presence of cholesterol in the system. To address the comment of the referee, in the supporting information we have now added a SAXS diffraction pattern of a Pn3m phase based on monopalmitolein with 25% cholesterol as Figure S6 and of Ia3d and

Im3m cubic phases formed by monopalmitolein with 5% of DSPG and 10% of cholesterol at 50% and 70% of water as Figure S7 and S8, respectively. Furthermore the sentence was re-edited to avoid possible misunderstanding:

“Although monopalmitolein supports the presence of cholesterol in the ensued mesophases, with and without DSPG (see supporting information Figures S6-S8), cholesterol was not added here since the membrane proteins targeted in the present work do not require its presence for reconstitution and crystallization.”

Below the three added figures.

REVIEWERS' COMMENTS:

Reviewer #2 (Remarks to the Author):

This is an important subject and a well organized paper proposing conditions for crystallizing large membrane proteins. As I mentioned in my first review, it deserves publishing after minor changes. The minor changes required have been introduced and the quality of this submission now is very high. The paper should be published in its present form and it will be interesting and cited by researchers involved in membrane protein studies and structural biology.

Reviewer #3 (Remarks to the Author):

The authors contend that they would likely be able to optimize the conditions to obtain an atomic resolution structure of GLIC. Given that GLIC has been solved by vapor diffusion at 2.95Å, I believe this is an important milestone to achieve before being considered for publication in Nature Communications. In my experience, it is not always possible to go beyond 6Å.